# Systematic identification of metabolites controlling gene expression in *E. coli*

Martin Lempp[1], Niklas Farke[1], Michelle Kuntz[1], Sven Andreas Freibert [2], Roland Lill [2,3] & Hannes Link [1,3]*

Metabolism controls gene expression through allosteric interactions between metabolites and transcription factors. These interactions are usually measured with in vitro assays, but there are no methods to identify them at a genome-scale in vivo. Here we show that dynamic transcriptome and metabolome data identify metabolites that control transcription factors in *E. coli*. By switching an *E. coli* culture between starvation and growth, we induce strong metabolite concentration changes and gene expression changes. Using Network Component Analysis we calculate the activities of 209 transcriptional regulators and correlate them with metabolites. This approach captures, for instance, the in vivo kinetics of CRP regulation by cyclic-AMP. By testing correlations between all pairs of transcription factors and metabolites, we predict putative effectors of 71 transcription factors, and validate five interactions in vitro. These results show that combining transcriptomics and metabolomics generates hypotheses about metabolism-transcription interactions that drive transitions between physiological states.

---

[1] Max Planck Institute for Terrestrial Microbiology, Marburg 35043, Germany. [2] Institut für Zytobiologie und Zytopathologie, Philipps-Universität Marburg, 35033 Marburg, Germany. [3] LOEWE Zentrum für Synthetische Mikrobiologie SYNMIKRO, Philipps-Universität Marburg, 35032 Marburg, Germany. *email: hannes.link@synmikro.mpi-marburg.mpg.de

Transcriptional regulation of metabolism is well characterized regarding the canonical flow of genetic information, which considers how transcription modulates the abundance of enzymes, and thereby metabolic flux and metabolites[1–4]. In reverse, metabolites convey information back to the transcription network by directly or indirectly interacting with a transcription factor (TF)[5–9] (Fig. 1a). In *Escherichia coli*, for example, the amino acid arginine allosterically regulates the activity of ArgR, which is a TF that controls genes involved in arginine biosynthesis, but the total regulon includes more than 400 genes[10]. Allosteric TF regulation allows a cell to tune gene expression depending on its metabolic state and theory shows that this regulation is robust and predictable by models[11]. An important consequence of allosteric TF regulation is that metabolites are not just biomass building blocks but they serve as internal signals with the potential to actively drive transitions between different physiological states.

It is largely unexplored which of the many intracellular metabolites interact with TFs[12,13], yet many transcriptional regulators are expected to bind a small molecule[5]. Currently, a major limitation to fill this gap of knowledge is the lack of methods to identify the most functionally relevant metabolite–TF interactions that control gene expression in vivo. Detection of physical interactions between metabolites with transcriptional regulators is mainly based on in vitro assays, which are low-throughput, feasible for only certain compounds and combinatorial effects

cannot be assayed[14]. An alternative approach is to probe protein structural changes with proteomics, which can detect binding of a single metabolite across thousands of proteins in cell extracts, but this approach cannot decipher unspecific binding from interactions that are functional in vivo[15]. An in vivo approach has been proposed, which searches for correlations in metabolomics data and data from fluorescent transcriptional reporters. This method could indeed recover few of the known metabolites that are relevant for gene regulation of central carbon metabolism in *E. coli*[16].

Here, we measure the *E. coli* transcriptome and metabolome changes during a 20 h dynamic transition, and show that integrating these two data-types generates hypotheses about metabolite–TF interactions that may have functional relevance in vivo. We also construct a metabolite–TF network for *E. coli* from the literature and databases, and show that our approach recovers more than 50% of the interactions in this network that were covered by our data. Moreover, we validate five predicted interactions with in vitro binding assays, i.e. lysine–ArgR, tyrosine–TrpR, glutamate–SgrR, tryptophan–SoxR, and dihydroxyacetone phosphate–DhaR, showing that our methodology generates physiologically meaningful results.

## Results

**Switching *E. coli* between growth and carbon starvation.** We used a 1 L bioreactor to switch the culture conditions of *E. coli*

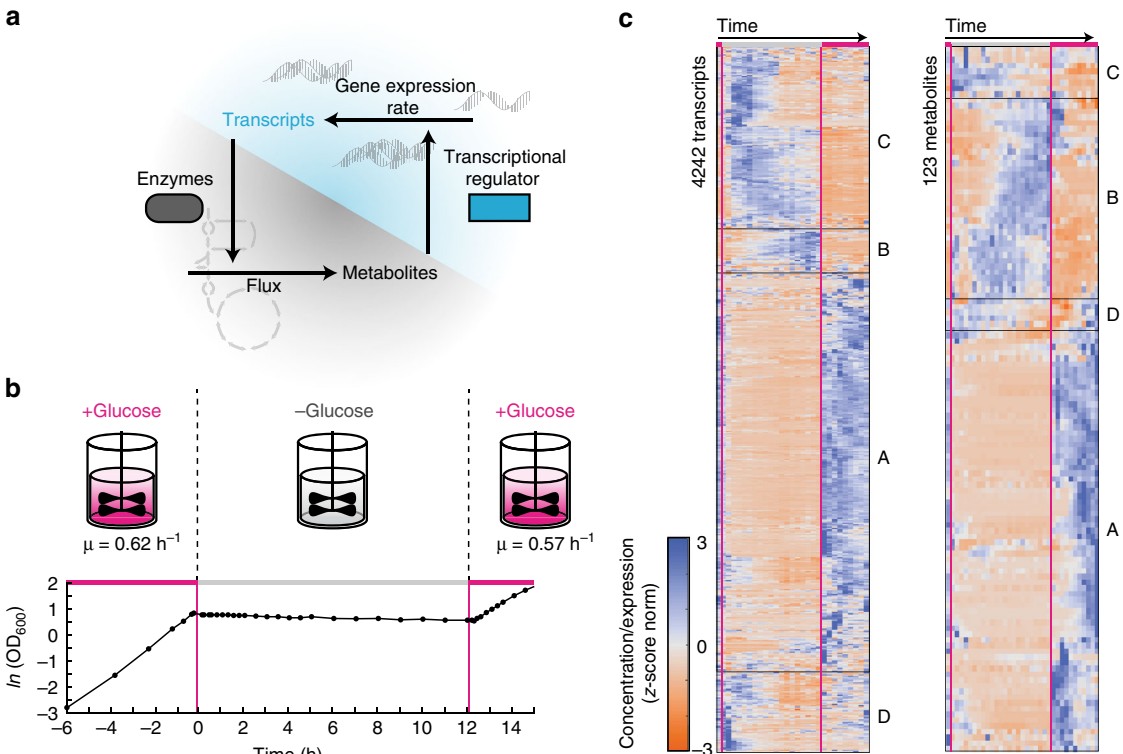

**Fig. 1** Dynamic metabolomics and transcriptomics during the growth–starvation switch in *E. coli*. **a** Schematic of the mutual feedback between metabolism and transcription. Transcription regulates enzyme levels, which affect flux and metabolite concentrations. Metabolite concentrations regulate gene expression by allosteric interactions with transcriptional regulators. **b** Growth of *E. coli* during the switch between growth, carbon starvation, and back to growth. Cells were cultivated in a 1 L bioreactor on glucose minimal medium to an OD of 2 and then the medium was exchanged to minimal medium without carbon source. After 12 h carbon starvation glucose was added to the culture. μ is the growth rate calculated by linear regression in the first and second growth phase. **c** Dynamic transcriptomics and metabolomics data measured at 29 and 35 time points, respectively. The first measurement was before the switch to starvation, 19 samples were collected during starvation and 9 samples during exit from starvation. Shown are z-score normalized transcript levels (in TPM) of 4242 genes and the z-score normalized concentration of 123 metabolites. Blue indicates high expression/concentration; orange indicates low expression/concentration. Data are grouped by hierarchical clustering and the four largest clusters are indicated as clusters A–D (average cluster dynamics are shown in Supplementary Fig. 2). Source data are provided as a Source Data file

between 6 h growth, 12 h carbon starvation, and 2 h growth resumption. First, cells grew on minimal medium with glucose and when the culture reached an optical density (OD) of 2, we transferred cells to minimal medium without carbon source. This rapid medium exchange caused an immediate growth arrest and cells starved for a period of 12 h (Fig. 1b). After 12 h we added again glucose to the culture and within 10 min cells resumed growing exponentially (Fig. 1b). Apart from the fast growth resumption, also oxygen uptake and $CO_2$-production increased rapidly upon glucose addition and reached the same rate as before starvation (Supplementary Fig. 1). Thus, physiological parameters like growth and respiration change in a fast and reversible fashion when E. coli cells enter and exit carbon starvation. Next, we investigated metabolism and transcription during the growth–starvation–growth switch, and measured the concentration of 123 metabolites by LC-MS/MS (Supplementary Fig. 2) and 4242 transcripts by RNA-sequencing (Fig. 1c, Supplementary Data 1 and 2). In total, we collected transcriptomics samples at 29 and metabolomics samples at 35 different time points in duplicates from a single bioreactor (Supplementary Data 3), with average errors of 18% for metabolites and 16% for transcripts. Only 8% of metabolites and 17% of the transcripts did not change significantly in either phase. To explore global dynamics of the metabolome and transcriptome data, we grouped each data set into four clusters (hierarchical clustering, z-score normalized). The clusters showed that the largest group of metabolites (63%) and transcripts (68%) decreased during the starvation phase and increased during the exit-phase (Cluster A, Supplementary Fig. 3). This group included intermediates in glycolysis like fructose-bisphosphate, dihydroxyacetone phosphate, and acetyl-CoA, as well as the nucleotide-triphosphates ATP and GTP (Fig. 2). Another group of metabolites and transcripts accumulated during the first 4–6 h into starvation, such as the amino acids lysine and phenylalanine that originate from degradation of proteins[17] (Cluster C, Supplementary Fig. 3). Similarly, accumulation of nucleotide derivatives like hypoxanthine was

presumably a consequence of RNA degradation (Fig. 2). These data indicate that starving E. coli cells catabolize RNA and proteins during the early phase of starvation, an interpretation that is consistent with the relatively high production of $CO_2$ in this phase (Supplementary Fig. 1), and also with the expression of genes involved in RNA, protein, and glycogen degradation processes (Supplementary Fig. 4). Notably, expression of genes in glycogen degradation preceded the expression of genes in RNA and protein degradation (Supplementary Fig. 4), confirming that glycogen functions as short-term energy storage. After switching cells back to glucose, 95% of the metabolites and 78% of transcripts reached the same steady-state levels that they had before the starvation phase. However, for many metabolites and transcripts, it took over 1 h until they reached a steady state, thus indicating extensive regulation during the exit phase.

**Integrating metabolomics and transcriptomics data**. To identify metabolites that are potential regulators of gene expression during the growth–starvation-growth switch, we searched for correlations between dynamics of metabolites and transcripts. Because metabolites modulate transcription through allosteric TF regulation, we sought to determine the activity of TFs and other regulators like $\sigma^{70}$ and $\sigma^S$. The relationship between transcriptional regulators and their target genes is well-characterized in E. coli, in the form of a transcription regulation network[18]. A well-mapped transcription regulation network allowed us to infer activities of transcriptional regulators from measured gene expression profiles using algorithms like network component analysis (NCA)[19,20]. The NCA algorithm estimates activity profiles of transcriptional regulators, which minimize the error between theoretical and measured gene expression profiles (for 2167 genes that are mapped to 209 transcriptional regulators in the E. coli transcription regulation network). In total, we performed 100 searches with the NCA algorithm, each with a different randomized initial condition, such that we obtained means and confidence intervals for activity profiles of the 209

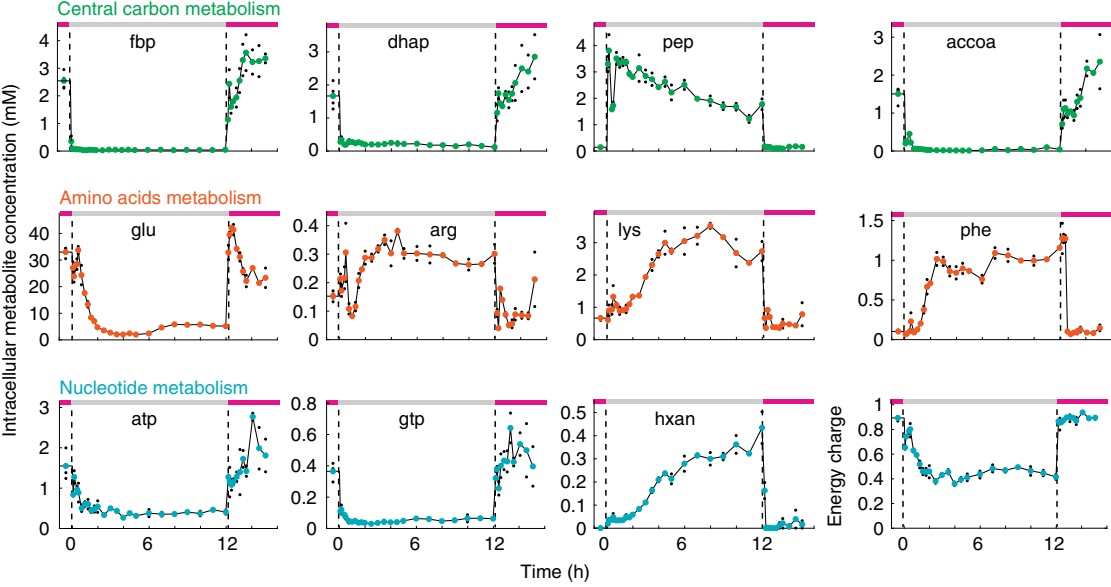

**Fig. 2** Examples of metabolite concentration changes during the growth–starvation-growth switch. Absolute concentration of metabolites in central carbon metabolism (green), amino acid metabolism (orange), and nucleotide metabolism (blue). The dashed lines indicate the starvation and growth phases. Black dots show concentrations of two replicates per time point (four at the first time point), colored dots are the mean. The energy charge is calculated from the concentration of ATP, ADP, and AMP. fbp fructose-1,6-bisphosphate, dhap dihydroxyacetone phosphate, pep phosphoenolpyruvate, accoa acetyl-coenzyme A, glu glutamate, arg arginine, lys lysine, phe phenylalanine, atp adenosine triphosphate, gtp guanosine triphosphate, hxan hypoxanthine. Source data are provided as a Source Data file

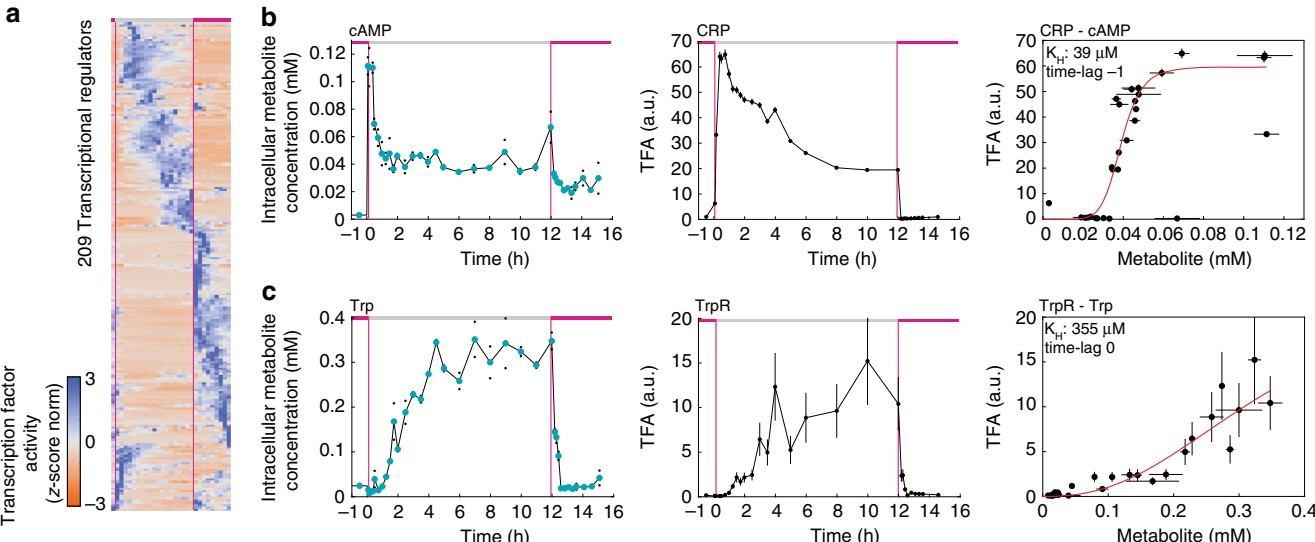

**Fig. 3** Metabolite levels and transcription factor activities recover regulatory interactions. **a** Z-score normalized activity of 209 transcriptional regulators. Blue indicates high activity; orange indicates low activity. Activity was estimated from the transcriptome data with network component analysis (NCA). **b** Dynamics of cyclic-AMP concentration (left) and CRP activity (middle) during the growth–starvation-growth switch. Correlation of cyclic-AMP and CRP activity (right). The red line is the best fit of a Hill function ($K_H = 39\,\mu M$). The correlation is shown with a time-lag of one data point to account for a time-delay between activity changes of CRP and changes in gene expression. **c** Same correlation analysis as in (**b**) for tryptophan (Trp) and the activity of TrpR, the repressor of the tryptophan operon. For cAMP and tryptophan, black dots show two replicates per time point, and colored dots are the mean. Error bars of transcription factor activity indicate the 95% confidence interval of $n = 100$ randomized estimates with NCA. Source data are provided as a Source Data file

transcriptional regulators (Fig. 3a, Supplementary Data 5). These 209 activity profiles were able to reproduce 75% of the transcript dynamics and were consistent with the expected responses of transcriptional regulators during starvation and growth[21]. For example, $\sigma^{70}$, the major sigma factor during exponential growth, was deactivated upon entry to starvation, and the stress response regulator $\sigma^S$ was immediately activated (Supplementary Fig. 5).

Allosteric regulation of a TF by a metabolite is often described by Hill-type kinetics[11], which assumes a sigmoidal relationship between TF activity and the concentration of an effector metabolite. In a canonical example of this regulation, the secondary messenger cyclic-AMP activates CRP, which is a global TF in *E. coli*[22,23]. On the basis of Hill kinetics, we tested how well the measured cyclic-AMP concentration predicts the activity profile of CRP (Fig. 3b). Cyclic-AMP and CRP activity revealed indeed a Hill-type relationship with an activation constant ($K_H$) of $39\,\mu M$, which is very close to the in vitro determined value of $27\,\mu M$[23]. Thus in vivo metabolite and transcript data identifies the existence of the known interaction between cyclic-AMP and CRP, and additionally captures the underlying kinetics of allosteric TF regulation. Another well-known metabolite–TF pair is tryptophan and the repressor of the tryptophan operon (TrpR), which also showed Hill-type kinetics, and the in vivo $K_H$ of $355\,\mu M$ was again relatively close to the in vitro value of $160\,\mu M$[24] (Fig. 3c).

Next, we wondered how many of the known metabolite–TF interactions are covered by our data, and whether they show a Hill-type relationship. Therefore, we first constructed a "literature network" of known metabolite–TF interactions by mining RegulonDB[18], the EcoCyc database[25] and the Allosteric Database[26]. This literature metabolite–TF network included in total 134 interactions between 87 TFs and 106 metabolites (Supplementary Fig. 6). 41% of the interactions are activating, 38% inhibiting and for 21% it is not known whether the metabolite inhibits or activates the TF. Our data covered interactions for 21 out of the 87 TFs, and 12 of them correlated with at least one of

the known regulatory metabolites (Fig. 4a, Pearson's correlation coefficient $R^2 > 0.75$). Thus, our data recovered known interactions in more than 50% of the cases, and in each of these cases the correlation correctly reflected, whether the metabolite activates or inhibits the TF. In case of NadR and ExuR, our data suggests that they are inhibited by ATP and lysine, respectively.

**Mapping metabolism-transcription interactions systematically**. A problem of the correlation analysis was that several metabolites correlated with the activity of a TF, resulting in many false positives (Fig. 4b). The large number of false positives is mainly caused by metabolites that have similar dynamics. The same problem was previously reported for a multi-omics analysis of yeast metabolism, which searched for correlations between metabolites and fluxes[27]. In this study, correlations between metabolites caused also many false positives, and including prior knowledge about metabolic flux regulation solved the problem. Here, we could not adapt such an approach, due to the limited information about allosteric TF regulation. Instead, we reduced the number of putative interactions by using a distance criterion for metabolites and TFs: metabolite–TF pairs were only considered, if at least one target-gene of the TF encodes an enzyme that participates in the same metabolic subsystem as the metabolite or if the metabolite is a substrate or a product. The hypothesis behind this distance criterion is that metabolites are more likely to regulate genes that are involved in their own biosynthesis. This assumption is supported by a recent study in cancer cells, which showed that metabolite-gene pairs have a higher correlation when they are close in the metabolic network[28]. We observed a similar proximity of metabolite–TF interactions in our literature network, because more than 80% of these interactions have a small distance in the *E. coli* genome-scale metabolic model[29] (Supplementary Fig. 6). We then applied the distance criterion to our data and only considered metabolite–TF pairs that fulfilled

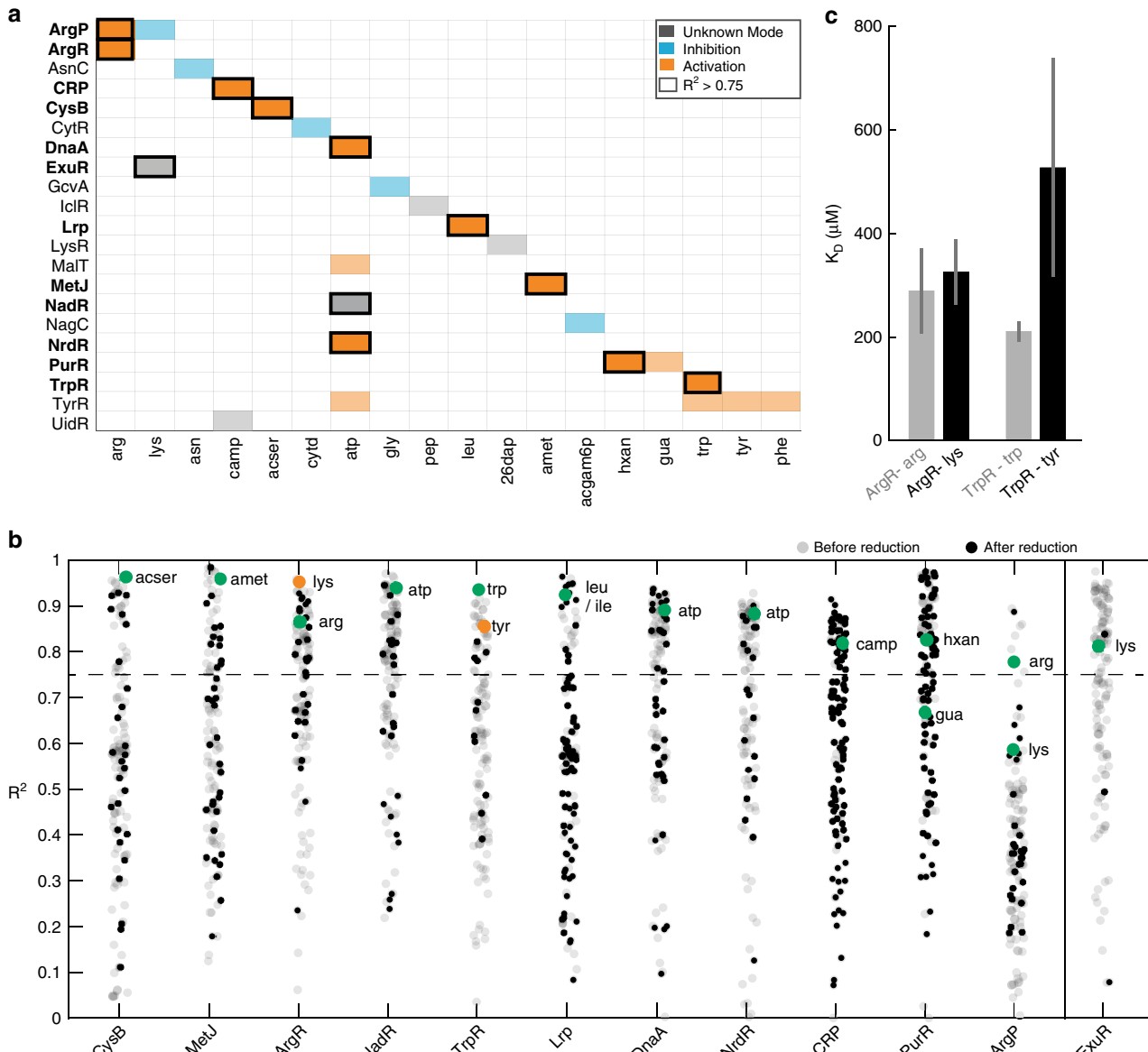

**Fig. 4** Identification and validation of known metabolite-transcription factor interactions. **a** Metabolite–TF interactions that are described in the literature and covered by the metabolome data in Fig. 1c and the TF activities in Fig. 3a. Shown are 21 transcription factors (rows) and their respective effector metabolites (columns). Orange indicates an activation of the TF by the metabolite, blue indicates an inhibition, and gray indicates that the mode is unknown. Metabolite–TF pairs that show Hill-type kinetics in the metabolome and transcriptome data ($R^2 > 0.75$) are indicated with a box. **b** Correlation coefficients of 12 transcription factors, which had activity profiles that correlated with at least one of the known effector metabolites (green dots). Gray and black dots are correlation coefficients with all other 123 measured metabolites. Black dots are metabolites that fulfill the distance criterion (same metabolic subsystem or product/substrate of a target-gene). Gray dots are metabolites that are rejected by the distance criterion. For ExuR the distance criterion excluded the known effector lysine. Lysine and tyrosine are indicated in orange for ArgR and TrpR. **c** In vitro measured dissociation constants ($K_D$) of ArgR and TrpR with the known effectors (arginine and tryptophan) and the predicted additional effectors (lysine and tyrosine). Binding was measured with His-tag purified ArgR and TrpR using micro scale thermophoresis (MST). Error bars show the 95% confidence interval of $K_D$ estimates, which are based on fitting $n = 9$ MST assays (proteins purified three times, each measured in three MST assays). MST data are shown Supplementary Fig. 9. arg arginine, lys lysine, asn aspartate, acser O-acetyl-Serine, cytd Cytidine, atp adenosine triphosphate, gly glycine, pep phosphoenolpyruvate, leu leucine, 26dap diaminopimelic acid, amet S-adenosylmethionine, acgam6p N-acetyl-glucosamine phosphate, hxan hypoxanthine, gua guanine, trp tryptophan, tyr tyrosine, phe phenylalanine. Source data are provided as a Source Data file

the distance criterion (black dots in Fig. 4b). For the 12 known metabolite–TF interactions that showed a Hill-type relationship, 11 fulfilled the distance criterion, and only the interaction between ExuR and lysine was rejected. The advantage of the distance filter was that it reduced the number of highly correlating metabolites from an average of 34 metabolites per TF to an average of 9 (Fig. 4b).

Among the false positives that remained after the distance filter were lysine–ArgR and tyrosine–TrpR (orange dots in Fig. 4b). Because lysine and tyrosine share structural similarity with the known allosteric effectors (arginine for ArgR and tryptophan for TrpR), we tested if lysine and tyrosine are additional and previously unidentified regulators of ArgR and TrpR. Therefore, we purified the two TFs and tested binding of lysine and tyrosine in vitro using

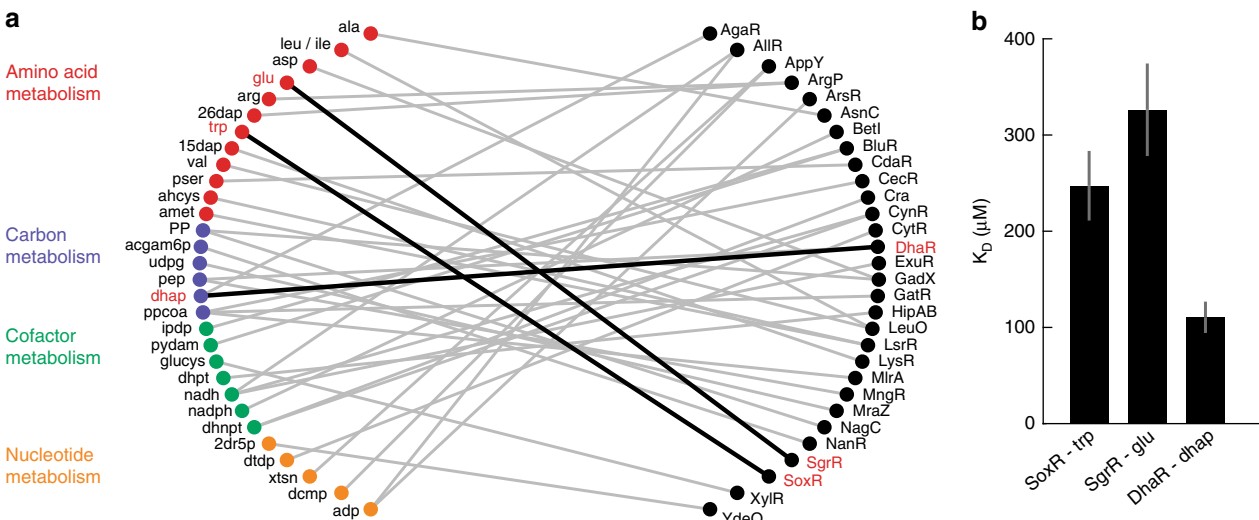

**Fig. 5** Identification and validation of new metabolite-transcription factor interactions. **a** Network of potentially new interactions between transcription factors and metabolites. Shown are 30 transcription factors that correlated with one or maximal two metabolites (only metabolites that fulfill the distance criterion). Metabolites are colored according to the subsystems of amino acid metabolism (red), carbon metabolism (purple), cofactor metabolism (green) or nucleotide metabolism (yellow). Connections in bold black highlight new interactions that are validated in vitro. **b** In vitro measured $K_D$ values of the new interactions indicated in bold in (**a**). Binding was measured with His-tag purified SoxR, SgrR, and DhaR using micro scale thermophoresis (MST). Error bars show the 95% confidence intervals of $K_D$ estimates, which are based on fitting $n = 9$ MST assays (proteins purified three times, each measured in three MST assays). MST data are shown Supplementary Fig. 9. ala alanine, leu/ile leucine/isoleucine, asp aspartate, glu glutamate, arg arginine, 26dap diaminopimelic acid, trp tryptophan, 15-dap 1,5-diaminopimelate, val valine, pser 3-phosphoserine, ahcys S-adenosylhomocysteine, amet S-adenosylmethionine, PP pentose phosphate, acgam6p N-acetyl-D-glucosamine-P, udpg UDP-glucose, pep phosphoenolpyruvate, dhap dihydroxyacetone phosphate, ppcoa propionyl-coenzyme A, ipdp isopentenyl diphosphate, pydam pyridoxamine, glucys gamma-glutamyl-cysteine, dhpt dihydropteroate, nadh nicotinamide adenine dinucleotide (reduced), nadph nicotinamide adenine dinucleotide phosphate (reduced), dhnpt dihydroneopterin, 2dr5p deoxyribose phosphate, dtdp thymidine diphosphate, xtsn xanthosine, dcmp deoxycytidine monophosphate, adp adenosine diphosphate. Source data are provided as a Source Data file

micro-scale thermophoresis (MST). The in vitro MST assays showed indeed binding of lysine and tyrosine to ArgR and TrpR, respectively, thus validating the in vivo prediction (Fig. 4c, Supplementary Fig. 9). The in vitro assays also confirmed the known arginine–ArgR and tryptophan–TrpR interactions (Fig. 4c, Supplementary Fig. 9). Because ArgR regulates essential steps in lysine biosynthesis, as well as two lysine transporters, the physiological function of the lysine–ArgR interaction is presumably a metabolic feedback that inhibits lysine production and import when lysine is abundant[30,31]. In case of TrpR, previous studies showed that deletion of TrpR, affects expression of *tyrA* in the tyrosine biosynthesis pathway[32]. Here, we show that also tyrosine is linked to TrpR, and the crosstalk between the two aromatic amino acids could potentially coordinate their biosynthesis.

Finally, we tested if we can generate hypotheses about the existence of metabolite–TF interactions in an unbiased fashion by fitting Hill functions to all pairs of metabolites and TFs. We first reduced the number of TFs from 209 to 125 by excluding: (i) TFs that followed simple on-off-on dynamics, (ii) TFs with poor estimates of activity profiles (confidence interval >100%), and (iii) TFs that are part of two-component systems (these regulators are more likely modulated by external signals rather than internal metabolites). The remaining 125 TFs and 123 metabolites resulted in 15,375 metabolite–TF pairs, for which we tested if they show a Hill-type relationship. A total of 3067 metabolite–TF pairs (20%) showed a Hill type relationship ($R^2 > 0.75$ Supplementary Data 6), and by applying again the distance criterion we reduced this number to 513, which we considered as putative metabolite–TF interactions (Supplementary Figs. 7, 8 and Supplementary Data 7).

The putative 513 interactions included 71 TFs, and we focused on the 30 TFs that correlated only with one or two metabolites (Supplementary Data 7). The resulting network shows mostly interactions of TFs with metabolites from amino acid and nucleotide metabolism but also with intermediates in carbon and cofactor metabolism (Fig. 5a). We purified three of the identified TFs to test if they bind the predicted metabolite. In vitro MST assays validated that SoxR binds tryptophan, SgrR binds glutamate, and DhaR binds the glycolysis intermediate dihydroxyacetone phosphate (DHAP) (Fig. 5b, Supplementary Fig. 9). SoxR is known to activate the expression of *aroF* and *tyrA*, which encode enzymes catalyzing the first step in the biosynthesis of all aromatic amino acids (*aroF*) and the tyrosine branch (*tyrA*)[33]. By binding tryptophan, SoxR could be part of a feedback regulation circuit in aromatic amino acid biosynthesis, which reduces expression of *aroF* and *tyrA* when tryptophan levels are high. SgrR activates *alaC* that encodes a transaminase that converts glutamate and pyruvate to alpha-ketoglutarate and alanine[34]. This transaminase accounts, together with a corresponding isoenzyme, for 90% of the catalytic activity for biosynthesis of alanine in *E. coli*[35]. As our in vivo data show an inhibition of SgrR by glutamate, low glutamate levels would upregulate *alaC*. Because low glutamate level brings the transamination reaction closer to thermodynamic equilibrium, an accompanying upregulation of *alaC* might provide the necessary enzymatic capacity[36]. The last new interaction is DhaR and DHAP, a regulator of dihydroxyacetone kinases, which seems to activate in response to increasing DHAP levels[37]. As DhaR activates the dihydroxyacetone kinases, the interaction could be part of a positive feedback loop.

## Discussion

In conclusion, data of the *E. coli* transcriptome and metabolome during a 20 h starvation–growth–starvation switch generated hypotheses about potential interactions between metabolites and TFs. The scale of this approach is the biggest advantage, because it

allows pair-wise testing of all TFs against all metabolites. Here, we provided a first proof-of-principle that the combination of transcriptomics and metabolomics has a great potential to identify metabolite–TF interactions at a metabolism-wide scale. To this end, we showed that many of the known metabolite–TF interactions were reflected by our data (e.g., cyclic AMP-CRP), and, therefore, that metabolite and gene expression data contain the information that is necessary to reconstruct metabolic-genetic networks. Moreover, we could validate five of the predicted metabolite–TF interactions with in vitro assays (lysine–ArgR, tyrosine–TrpR, glutamate–SgrR, tryptophan–SoxR and dihydroxyacetone phosphate–DhaR).

In our analysis, we excluded two-component systems, because they are likely responsive to external metabolites. By measuring the exo-metabolome it should be possible to identify effectors of two-component systems with the method proposed in this study. We also excluded TFs with poor estimates of activity profiles, and to include these TFs one could probe their activities with fluorescent transcriptional reporters as recently suggested[16]. Accurate information about TF activities was important for our approach because it allowed pairwise testing of Hill-type relationships between TF activities and metabolites. Here, we inferred TF activities with the NCA algorithm that requires a well-mapped transcription regulation network. While the transcription regulation network is known in E. coli, it is unknown for most other organisms. To overcome the need for a known transcription regulation network, the TF activities could be inferred from the transcriptome data directly without using prior knowledge about the transcription regulation network. Previous studies showed for example that machine learning methods can infer TF activities in E. coli based on transcriptomics data[38], and inference of regulatory metabolites with such methods was also suggested[39]. Future approaches could even consider determining TF activities and regulatory metabolites simultaneously.

The main limitation in our study was that many metabolites showed similar dynamics, which in turn caused false positive predictions of metabolite–TF interactions. The high correlation among metabolites could be a general problem in metabolomics-based inference approaches[27]. A solution for this problem is to enforce more specific metabolite concentration changes by localized perturbations of metabolism, for example by disturbing single enzymes. We anticipate that the transcriptome and metabolome of hundreds of locally perturbed metabolic states would provide sufficient information to faithfully map metabolite–TF interactions of an organism. An effective perturbation method is CRISPR interference, because of its potential to interfere with the expression of every enzyme of an organism.

A complete map of metabolite–TF interactions would advance our knowledge about the dynamic nature of metabolic regulation and enable the construction of dynamic metabolic models. Here, we focused mainly on interactions that are part of metabolic-genetic feedback circuits, because we considered the distance between TFs and metabolites. However, metabolites will not only affect the transcription of genes encoding enzymes, but also affect genes involved in various other physiological processes. Understanding these long-ranging metabolite–TF interactions would dramatically increase our understanding about how metabolism drives physiological responses, e.g. to oxidative stress[40] or antibiotics[41]. Finally, there is the possibility to exploit the knowledge about metabolite–TF interactions to engineer better strains for biotechnology, e.g. by designing genetic-metabolic feedback that acts as valves in production strains[42] or growth switches[43].

## Methods

**Strains and cultivation.** E. coli BW25113 (parent strain for the Keio Collection, CGSC#: 7636) was cultivated in 1 L bioreactor with 500 mL of M9 minimal medium containing 5 g L$^{-1}$ glucose to an optical density at 600 nm (OD) of 2. Then the culture was centrifuged at 37 °C and 1800 × g for 5 min Pelleted cells were resuspended in M9 medium at 37 °C without glucose and transferred back to the bioreactor. After 12 h, the culture was supplemented glucose to a final concertation of 5 g L$^{-1}$ glucose. The M9 minimal medium consisted of the following components (per liter): 6 g Na$_2$HPO$_4$ · 2 H$_2$O, 3 g KH$_2$PO$_4$, 1.5 g (NH$_4$)$_2$SO$_4$, 0.5 g NaCl. The following components were sterilized separately and then added to the medium (final concentrations): 0.1 mM CaCl$_2$, 1 mM MgSO$_4$, 60 µM FeCl$_3$, 2.8 µM thiamine-HCl, and 10 mL trace salt solution. The trace salt solution contained (per liter) 180 mg ZnSO$_4$ · 7 H$_2$O, 120 mg CuCl$_2$ · 2 H$_2$O, 120 mg MnSO$_4$· H$_2$O, 180 mg CoCl$_2$ · 6 H$_2$O. The dissolved oxygen in the bioreactor was kept at 30% and pH 7 was controlled with 5 M NH$_4$OH and 20% H$_3$PO$_4$. The bioreactor was a BioFlo115 bioprocess system (Eppendorf, Hamburg, Germany), equipped with a pH-sensor (Mettler Toledo, Colombus, OH) and a DO-sensor (Mettler Toledo, Colombus, OH). Exhaust gas of the cultivation was analyzed by a DASGIP GasAnalyser (Eppendorf, Hamburg, Germany). The GasAnalyser was calibrated with two-point-calibration prior to the cultivation. The bioreactor cultivation was monitored with the BioCommand-Software (Eppendorf, Hamburg, Germany).

**Metabolomics.** For metabolomics 1 mL culture aliquots were vacuum-filtered on a 0.45 µm pore size filter (HVLP02500, Merck Millipore). Filters were immediately transferred into 40:40:20 (v-%) acetonitrile/methanol/water at −20 °C for extraction. Extracts were centrifuged for 15 min at 11,000 × g at −9 °C. Centrifuged extracts were mixed with $^{13}$C-labeled internal standard. Chromatographic separations were performed on an Agilent 1290 Infinity II LC System (Agilent Technologies) equipped with an Acquity UPLC BEH Amide column (2.1 × 30 mm, particle size 1.8 µm, Waters) for acidic conditions and an iHilic-Fusion (P) HPLC column (2.1 × 50 mm, particle size 5 µm, Hilicon) for basic conditions. We were applying the following binary gradients at a flow rate of 400 µl min$^{-1}$: acidic condition) 0–1.3 min: isocratic 10% A (water/formic acid, 99.9/0.1 (v/v), 10 mM ammonium formate), 90% B (acetonitrile/formic acid, 99.9/0.1 (v/v)); 1.3–1.5 min linear from 90 to 40% B; 1.5–1.7 min linear from 40 to 90% B, 1.7–2 min isocratic 90% B. Basic condition) 0–1.3 min: isocratic 10% A (water/formic acid, 99.8/0.2 (v/v), 10 mM ammonium carbonate), 90% B (acetonitrile); 1.3–1.5 min linear from 90 to 40% B; 1.5–1.7 min linear from 40 to 90% B, 1.7–2 min isocratic 90% B. The injection volume was 3.0 µl (full loop injection).

Eluting compounds were detected using an Agilent 6495 triple quadrupole mass spectrometer (Agilent Technologies) equipped with an Agilent Jet Stream electrospray ion source in positive and negative ion mode. Source gas temperature was set to 200 °C, with 14 L min$^{-1}$ drying gas and a nebulizer pressure of 24 psi. Sheath gas temperature was set to 300 °C and flow to 11 L min$^{-1}$. Electrospray nozzle and capillary voltages were set to 500 and 2500 V, respectively. Metabolites were identified by multiple reaction monitoring (MRM), and MRM parameters were optimized and validated with authentic standards[44]. Metabolites were measured in $^{12}$C− and $^{13}$C isoforms, and data were analyzed with published Matlab code[44]. Metabolites were sampled four times at the first time point $t_0$; and two samples were collected at the remaining time points (see also reporting standards in Supplementary Data 10). Metabolomics metadata is accessible under the MetaboLights accession number MTBLS1044.

**Transcriptomics.** For transcriptomics 0.5 mL culture was transferred into reaction tubes and centrifuged at 11.000 × g for 2 min, and the pellet was frozen in liquid nitrogen. The total RNA of the cells was isolated using the Total RNA Isolation Mini Kit (Agilent, Santa Clara, CA). The integrity of the RNA was measured using the BioAnalyzer Pico-Kit (Agilent, Santa Clara, CA). RNA-sequencing was performed by the Max Planck-Genome-Centre Cologne, Germany (https://mpgc. mpipz.mpg.de/home/). The sequencing reads were analyzed and mapped using the CLC Software (QIAGEN, Venlo, NL). For normalization, gene expression was calculated as transcripts per kilobase million (TPMs). RNA was sampled four times at the first time point $t_0$; and two samples were collected at the remaining time points. For the time points $t_{13}$, $t_{15}$, $t_{19}$ and $t_{24}$ one of the two replicates was excluded due to low quality of the sampled RNA. Transcriptomics metadata is accessible under the GEO number GSE131992.

**Network component analysis (NCA).** NCA was performed by iteratively optimizing connectivity strength and TF-activity by using the connectivity matrix of the transcription regulation network and the measured gene expression. The optimization is a least square optimization between the gene expression and the product of connectivity and TF-activity:

$$\min_{A,P} \|E - AP\|^2 \qquad (1)$$

Where $E$ is the log$_{10}$ transformed gene expression data (in TPMs) (Supplementary Data 9), $A$ the connectivity matrix of the transcription regulation network (matrix with regulator-gene interactions Supplementary Data 8) and $P$ the TF-activity[19]. To generate the connectivity matrix, a matrix of transcription regulator—gene interactions was generated by combining the matrixes of TF— gene interactions and sigma factor—gene interactions of RegulonDB[18]. Additional regulation that was added was the (p)ppGpp regulon and transcriptional attenuation, as described in the EcoCyc database[25]. To account for basal expression of every gene by the RNA

polymerase we added a global regulator, which was connected to all genes in the connectivity matrix. Randomized starting points were used for each calculation cycle of the algorithm. A calculation cycle was aborted if the summed squared 2-norm of the residuals did not change by more than 1%.

**Correlations between metabolites and TF activities**. Metabolite concentrations and TF activities were first correlated linearly. In case of a positive linear correlation, we used activating Hill kinetics as the basis for a non-linear fit. In case of a negative linear correlation we used inhibition kinetics:

$$Activation\ kinetics: y = y_{max} * \frac{x^h}{x^h + K_H^h} \tag{2}$$

$$Inhibition\ kinetics: y = y_{max} * \frac{K_H^h}{x^h + K_H^h} \tag{3}$$

Where $y$ is the TF activity, and $x$ the metabolite concentration. $K_H$ is the activation constant, $h$ the Hill coefficient and $y_{max}$ is the maximal TF activity, which was assumed to be constant over time. Parameters of the Hill equations ($K_H$ and $h$) were estimated in total 50 times per metabolite–TF pair. The Hill coefficient $h$ was constrained to an upper value of 10. For each pair of metabolite and TF, we tested if a negative time-shift of the TF activity by one time point would improve the parameter estimation. This accounts for the fact that TF activities are derived from gene expression data, which could potentially succeed changes of metabolite levels (Supplementary Fig. 10). The correlation coefficient $R^2$ was calculated between the measured TF activity and the transformed metabolite levels using the estimated Hill parameters.

**Distances of metabolite–TF interactions**. First, we remove all cofactors, as well as periplasmatic and extracellular metabolites from the stoichiometric matrix of the iJO1366 metabolic genome-scale model of *E. coli*. Next, we create a metabolite-gene adjacency matrix, F, by calculating the inner product of the modified stoichiometric matrix, N, and the reaction-gene matrix, G. We finish by computing the Boolean of F, F'. Next, we transform F' it into an undirected, bipartite graph, nodes denoting metabolites and genes, respectively. For this graph, we calculate a distance matrix, D, containing all pairwise distances between metabolites and genes in F. For known metabolite–TF interactions, we look for the distances between the regulating metabolite and each of the target genes of the TF and take the smallest distance. In case a regulating metabolite is not part of the iJO1366, we omit the distance calculation[45].

The distance criterion for correlating metabolite–TF pairs (Fig. 4b) was also based on the genome-scale model iJO1366[29]. Pairs of metabolites and TFs were only considered if at least one of the two criteria was fulfilled. Criteria 1: the metabolite is a product or a substrate of an enzyme that is encoded by a target-gene of the TF. Criteria 2: the metabolite is listed in the same metabolic subsystem as an enzyme that is encoded by a target-gene of the TF. Subsystems of TFs were defined as the metabolic pathways controlled by the TF in the genome scale model. Subsystems of metabolites were defined according to the Supplementary Data 1.

**Protein overexpression and purification**. TFs were purified from the *E. coli* ASKA strains[46]. Cells were grown in 200 mL TB medium containing 30 μg × mL$^{-1}$ chloramphenicol at 37 °C. When cells reached OD 0.6 we added 0.5 mM IPTG. Cells were incubated at 37 °C for 3 h more and harvested by centrifugation. Proteins were purified from the pellets using Protino™ Ni-TED-IDA 1000 Kit (Macherey-Nagel, Düren Germany). Protein purity was confirmed by SDS-PAGE and concentrations were determined by the Pierce protein BCA Assay (Thermo Fischer Scientific, Waltham, MA).

**Quantitation of interactions by microscale thermophoresis**. Microscale Thermophoresis (MST)[47] was performed on a Monolith NT.115 (Nano Temper Technologies GmbH, Munich, Germany) at 21 °C (red LED power was set to 75% and infrared laser power to 80%). 50 nM of the respective protein was labeled with the dye Monolith His-Tag Labeling Kit RED-tris-NTA 2nd Generation (MO-L018) supplied by NanoTemper Technologies. Labeled proteins were titrated as indicated with the respective metabolite in buffer T (50 mM NaH$_2$PO$_4$, 500 mM NaCl, and pH 5.7). At least nine independent MST experiments (three technical replicates of three biological replicates) were performed at 680 nm and processed by Nano Temper Analysis package 1.2.009 and Origin8 (OriginLab, Northampton, MA).

**Reporting summary**. Further information on experimental design is available in the Nature Research Reporting Summary linked to this paper.

## Code availability

Matlab code to perform Network Component Analysis and Kinetic correlations can be accessed from the GitHub repository via https://github.com/nfarke/Lempp_Metabolite_TF_interaction_Ecoli.

## Data availability

Gene expression data that support the findings of this study have been deposited in NCBI's Gene Expression Omnibus with the accession code GSE131992. Metabolomics data that support the findings of this study have been deposited in MetaboLights database with the accession codes MTBLS1044. The source data of Figs. 1a, b, 2, 3a–c, 4b, c and 5b and Supplementary Figs. 1, 3, 4, 5, 7, and 9 are provided as a Source Data file. All other data are available from the corresponding author on reasonable request.

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

## Acknowledgements

We thank G. Bange and K. Drescher for discussions. This work was supported by the Deutsche Forschungsgemeinschaft through the Collaborative Research Center SFB 987, and by the ERC starting grant 715650. M.K. acknowledges founding of the IMPRS graduate school for environmental, cellular and molecular microbiology from the Max Planck Society. We thank the Max Planck-Genome-Centre Cologne (http://mpgc.mpipz.mpg.de/home/) for performing RNA sequencing in this study.

## Author contributions

M.L. and M.K. performed experiments. M.L. performed LC-MS/MS measurement, Network Component Analysis, kinetic correlations. M.L., S.F. and R.L. performed MST. N.F. constructed and analyzed the literature metabolite-TF network. M.L. and H.L. co-wrote the paper. H.L. directed the project.

## Competing interests

The authors declare no competing interests.

## Additional information

**Peer Review Information** *Nature Communications* thanks Julio Collado-Vides, Alisdair Fernie and the other, anonymous, reviewer(s) for their contribution to the peer review of this work. Peer reviewer reports are available.

