## [Peer Review File · Nature Communications]

Reviewers' comments:

Reviewer #1 (Remarks to the Author):

Lempp et al. present a systematic study for investigating metabolite regulated gene expression in E.coli. It is a tour-de-force of the approach and not only does it use the broad-brush approach to identify potentially regulations but validates them with high quality biochemistry of metabolite transcription factor interactions. To the best of my knowledge this is a highly novel approach and apparently a powerful one too. The only minor suggestion I would state is that the scope should be better developed at the end of the paper in order to better describe to the reader the massive potential of this approach.

In addition I feel that the authors should provide metadata concerning their metabolomics with several suggested reporting standards present in the literature which provide good examples of what should be provided.

Reviewer #2 (Remarks to the Author):

This is a well written manuscript, describing the implementation of a systematic approach using transcriptomic and metabolomic time course measurements in order to identify a large number of metabolites that control gene expression allosterically binding to transcription factors (TFs) in E.coli. A major claim of the paper is that these interactions are mostly based on in vitro low-throughput experiments, and they present a genome-scale in vivo approach. As a result potential effectors are identified for 65 TFs in E.coli. No doubt, if true, this is a potentially relevant contribution to enrich the battery of high-throughput approaches to accelerate our deciphering of the genetic machineries of different genomes governing changes in gene expression and their interplay with metabolism.

However, there are several problems that need to be addressed to support the major claim of the paper in a more convincing way. First, the authors do not discuss at all the limitations of the method. I suggest they present an organized discussion of the limitations and the corresponding alternatives to address them, if this is really a genome-scale approach. Second, there is no emphasis on validation. Given the considerable amount of available knowledge of the E.coli network, the authors should describe the fraction of known interactions that were found by this approach. We do not

even know how many of the 65 TF-metabolite interactions here identified are novel and how many are part of the 40 already known in E.coli.

Finally, in addition to the methodological novelty, the data is valuable, but is not presented properly for easy use by the community. It is a pity that only two repetitions were performed, and in fact in some few time points only one measurement was used.

Below I get into more detail of these major issues, and finally add a list of minor concerns.

Limits to the method and validation.

The E.coli network is estimated to comprise around 300 TFs, but they input only 209 in the network component analysis (NCA). Is NCA limited only to TFs with a priori known target genes? If so, this is a major limitation to apply this approach in other genomes, where the regulatory network is much less characterized, or essentially unknown. I wonder if the dynamics of autoregulatory interactions, rather common in TFs, may pose difficulties to the reverse engineering within the NCA method.

Out of 209, after filtering, the final result is the identification of potential allosteric effectors for 65 TFs. It is almost logically mandatory to discuss somewhere, what alternative are there to identify allosteric metabolites for the remaining 144 TFs.

I do not see the logic to exclude two component systems. Many TFs have as signals external metabolites, so if that is the case of the two-component systems, it is not a unique feature of that class. Is NCA, or the experimental setting, that is limited to search for correlations with internal metabolites? This is relevant given the large fraction of external sensing of the regulatory network machinery.

The approach relies on NCA, where transcription factor activity is an essential concept, nonetheless there is no explanation in the manuscript, how was TF activity obtained. It must have been inferred precisely using NCA, but this is said in a confusing way: “..we first calculated the activity of transcription factors and other regulators like sigma70 and sigmaS. To infer the activities of all transcriptional regulators in E.coli, we used NCA..”

In order to identify the TFs that correspond to the metabolites, the authors fitted Hill functions to all pairs of metabolites and TFs, generating 3990 potential interactions. These were then filtered, first eliminating all TFs that followed a simple on-off dynamics which reduced the potential interactions to close to 3000. Then metabolic feedback circuits were searched by calculation the distance in the metabolic network between the metabolite and the genes regulated by the TF. This brought potential interactions down to 336 and 65 TFs.

There is absolutely no mention on how such a distance was measured. Second, I suggest the authors should show if this distance is indeed small for the known allosteric metabolites of the E.coli network, to strengthen their claim.

I did not understand the addition of “a global generic regulator” to the connectivity matrix. Which global regulator is that? And what do we learn from this?

The value of the data.

Table S6 has the correlation coefficient for Hill-type kinetics for all pairs of metabolites and TFs. I suggest the authors put in bold those coefficients that correspond to the 336 filtered interactions, given the quote to that table in the manuscript, and the relevance of their filtering. Additionally, a new table should be added listing the final set of identified potential TF-metabolite interactions.

More detailed observations.

The E.coli BW25113 strain was used. The authors did not mention if they checked consistency in the use of databases used for the known metabolites with this particular strain.

In the first sentence of page 3, the authors refer to Fig 3b, when they mean Fig 3a.

Figure S6 could be better changed the X scale to millimolar.

The description in methods of the conditions of growth does not seem to me the most appropriate for the sake of easy comparison with other experiments in the literature. I think it should describe the molar or millimolar concentration of each component, instead of describing the amounts added to 1 liter, such as “1 mL of 0.1M CaCl₂”.

Finally, I should say that given the dependency on the NCA method, and other other features of this work, it is my appreciation that it is not easy for a researcher to reproduce this work, or even the analysis from the raw data. I understand this is not a simple issue, I suggest the authors to think if there are means to enhance the reproducibility of the work somehow.

Reviewer #3 (Remarks to the Author):

Lempp et al. propose a strategy for systematic identification of metabolites that control transcription through transcription factors by integrating dynamic metabolomics and transcriptomics measurements. The proposed method was used to both validate many known metabolite-transcription factor interactions in *E. coli* and to predict a significant number of potential new interactions. A small number of the new interactions were validated experimentally in vitro. The manuscript is overall well written and clear and the figures are informative.

Major comments:

1) The network component analysis (NCA) approach the authors use to infer transcription factor activities from global transcriptome profiles has been previously shown to be useful, but also suffers from some clear limitations including the requirement to a priori specify all the TF-target gene interactions that will be considered and the specific model of how multiple TFs acting on the same target promoter are assumed to contribute to the expression of the target gene. I have multiple suggestions regarding the NCA part:

a. The manuscript does not clearly state these underlying assumptions when the NCA results are described (page 3, middle paragraph). The major assumptions should be described here to inform the read better of what NCA can and what it can not do.

b. It would be informative to show the NCA inferred TF activities in Figure 1 in addition or instead of showing all the transcript levels as the TF activity data is actually what is used in all the other analyses.

c. If I understood it correctly the authors did not allow time lags between TF activity and transcription of its target genes (on the other hand they did consider time lags between metabolite levels and TF activity). Did the authors consider including time lags in the analysis or at least check whether considering time lags could have influenced the results?

2) As the authors themselves clearly state, the major weakness of the proposed approach is its high false positive metabolite-TF interaction prediction rate. This appears to be primarily driven by the specific experimental conditions used in the study where many metabolites and transcripts follow switch-like on-off dynamics as a function of the switch between non-starved and starved conditions. This forces the authors to filter the list of potential interactions using multiple criteria some of which are better justified than others. I have again several suggestions for clarifying/improving this part:

a. The authors a priori exclude two component systems from the analysis as these are not supposed to be modulated by internal metabolite levels. However, it would have been interesting to

see as a kind of a negative control how many metabolite-TF interactions would have been inferred for two component systems if they were included and if they indeed would have had fewer inferred interactions than other types of TFs. These types of controls are important for determining how false positive prone the overall approach is.

b. The second exclusion criterion the authors used was to exclude any TF that significantly correlated with simple on-off pattern (depending on the starvation condition). It could be interesting to try a different approach to this where specific metabolite-TF interactions would be included only if the metabolite-TF correlation was higher than the on-off pattern correlation.

c. The strongest exclusion criterion used by the authors was requiring that the metabolite and TF target genes are part of the same subsystem and the distance from metabolite to a target gene of the TF is less than two steps in the metabolic network. This step is not described well enough in the methods (i.e. what is the definition of a subsystem and what is meant by a step exactly). Also, it would be interesting to show the R^2 in Fig. 4a as a function of the metabolic distance used as a filtering criterion before the filtering (i.e. considering distances larger than 2 also). Finally, the authors should provide detailed information in supplementary material on exactly how many interactions were excluded based on the subsystem and distance criteria for each subsystem.

Minor comments:

1) Some additional interesting controls to check:

a. Did the energy charge correlate with any TF activity?

b. Did the “global regulator” level correlate with any metabolite level?

2) Page 8. Normalizing by TPM is not considered to be a terribly good approach nowadays. Consider using alternative better normalization methods in future studies and potentially also check for this study whether there is any significant effect due to the particular normalization used. There are many good assessment studies of different RNA-seq normalization methods. This could also improve the reproducibility of the data.

3) Page 8. The authors state that the Hill coefficients were “constrained to 10” – does this mean that they were constrained to be less than 10.

4) Page 9. “puffer” -> “buffer”

We thank all reviewers for their constructive comments, on which we based new analysis of our data and revisions of the manuscript. A point by point response is below and the most important changes are:

- **Metadata of Transcriptomics and Metabolomics** are now available from the following online repositories
Transcriptomics: GEO accession number GSE131992
<https://www.ncbi.nlm.nih.gov/geo/query/acc.cgi?acc=GSE131992>
Metabolomics: MetaboLights study identifier MTBLS1044
<https://www.ebi.ac.uk/metabolights/reviewer63fe3fb6-38c6-4033-8118-7e9409658376>
<https://www.ebi.ac.uk/metabolights/MTBLS1044> (release date: August 15th 2019)
- To estimate the success rate of our approach we mined the literature and databases for known metabolite-transcription factor interactions. We then compare this **“literature” metabolite-TF network** with our data (Fig. 4a). More than 50% of the known interactions were recovered by our approach.
- We provide a detailed **discussion** about the limitations of the approach, and describe its potential applications in mapping metabolic regulation.
- We performed all **additional data analysis** as suggested by the reviewers:
 - We tested if two component systems correlate with fewer metabolites
 - We compared the distance and the correlation of each metabolite-TF pair
 - We tested an alternative approach to filter on-off dynamics

Response to Reviewer 1

Lempp et al. present a systematic study for investigating metabolite regulated gene expression in E. coli. It is a tour-de-force of the approach and not only does it use the broad-brush approach to identify potentially regulations but validates them with high quality biochemistry of metabolite transcription factor interactions. To the best of my knowledge this is a highly novel approach and apparently a powerful one too. The only minor suggestion I would state is that the scope should be better developed at the end of the paper in order to better describe to the reader the massive potential of this approach.

We thank this reviewer for the positive and helpful comments. In the revised manuscript we added a more detailed discussion that defines the scope of the approach and possible applications in basic biology and biotechnology.

“A complete map of metabolite-TF interactions would advance our knowledge about the dynamic nature of metabolic regulation and enable the construction of dynamic metabolic models. Here, we focused mainly on interactions that are part of metabolic-genetic feedback circuits, because we considered the distance between TFs and metabolites. However, metabolites will not only affect transcription of genes encoding enzymes, but also affect genes involved in various other physiological processes. Understanding these long-ranging metabolite-TF interactions would dramatically increase our understanding about how metabolism drives physiological responses, e.g. to oxidative stress⁴⁰ or antibiotics⁴¹. Finally, there is the possibility to exploit the knowledge about metabolite-TF interaction to engineer better strains for biotechnology, e.g. by designing genetic-metabolic feedback that acts as valves in production strains⁴² or growth switches⁴³.”

In addition I feel that the authors should provide metadata concerning their metabolomics with several suggested reporting standards present in the literature which provide good examples of what should be provided.

We published our transcriptomics and metabolomics metadata on online repositories (MetaboLights and GEO). Additionally, we provide all source data in supplementary tables 1 and 2, as well as metabolomics reporting standards in table S10. We added this in the revised manuscript:

“Gene expression data that support the findings of this study have been deposited in NCBI’s Gene Expression Omnibus with the accession code GSE131992 (<https://www.ncbi.nlm.nih.gov/geo/query/acc.cgi?acc=GSE131992>). Metabolomics data that support the findings of this study have been deposited in MetaboLights database with the accession codes MTBLS1044 (<https://www.ebi.ac.uk/metabolights/MTBLS1044>). The source data of Figures 1a-b, 2, 3a-c, 4b-c and 5b and Supplementary Figures 1, 3, 4, 5, 7 and 9 are provided as a Source Data file.”

Response to Reviewer 2

This is a well written manuscript, describing the implementation of a systematic approach using transcriptomic and metabolomic time course measurements in order to identify a large number of metabolites that control gene expression allosterically binding to transcription factors (TFs) in *E. coli*. A major claim of the paper is that these interactions are mostly based on in vitro low-throughput experiments, and they present a genome-scale in vivo approach. As a result potential effectors are identified for 65 TFs in *E. coli*. No doubt, if true, this is a potentially relevant contribution to enrich the battery of high-throughput approaches to accelerate our deciphering of the genetic machineries of different genomes governing changes in gene expression and their interplay with metabolism.

However, there are several problems that need to be addressed to support the major claim of the paper in a more convincing way. First, the authors do not discuss at all the limitations of the method. I suggest they present an organized discussion of the limitations and the corresponding alternatives to address them, if this is really a genome-scale approach.

We thank the reviewer for the positive and helpful comments. As suggested we provide a more detailed discussion in the revised manuscript, which describes current limitations of the approach and strategies to overcome them.

One limitation is the need for a well-mapped transcription regulation network:

*“Accurate information about TF activities was important for our approach because it allowed pairwise testing of Hill-type relationships between TF activities and metabolites. Here, we inferred TF activities with the NCA algorithm that requires a well-mapped transcription regulation network. While the transcription regulation network is known in *E. coli*, it is unknown for most other organisms. To overcome the need for a known transcription regulation network, the TF activities could be inferred from the transcriptome data directly without using prior knowledge about the transcription regulation network. Previous studies showed for example that machine learning methods can infer TF activities in *E. coli* based on transcriptomics data³⁸, and inference of regulatory metabolites with such methods was also suggested³⁹. Future approaches could even consider determining TF activities and regulatory metabolites simultaneously. “*

Another limitation is the number of false positives due to correlations among metabolites:

“The main limitation in our study was that many metabolites showed similar dynamics, which in turn caused false positive predictions of metabolite-TF interactions. The high correlation among metabolites could be a general problem in metabolomics-based inference approaches²⁷. A solution for this problem is to enforce more specific metabolite concentration changes by localized perturbations of metabolism, for example by disturbing single enzymes. We anticipate that the transcriptome and metabolome of hundreds of locally perturbed metabolic states would provide sufficient information to faithfully map metabolite-TF interactions of an organism. An effective perturbation method is CRISPR interference, because of its potential to interfere with the expression of every enzyme of an organism.”

Second, there is no emphasis on validation. Given the considerable amount of available knowledge of the E.coli network, the authors should describe the fraction of known interactions that were found by this approach. We do not even know how many of the 65 TF-metabolite interactions here identified are novel and how many are part of the 40 already known in E. coli.

We agree with this point and tested systematically how many of the already known interactions were found by our approach. Therefore, we first created a network of the currently known metabolite-TF interactions in *E. coli*, based on RegulonDB, Ecocyc and the Allosteric Database. This network is shown in a new Figure S6 and the part which was covered by our data is shown in Figure 4a. In this network we recovered interactions for 12 out of the 21 transcription factors. We highlighted in the revised Table S7 the known interactions and potentially new interactions.

Finally, in addition to the methodological novelty, the data is valuable, but is not presented properly for easy use by the community. It is a pity that only two repetitions were performed, and in fact in some few time points only one measurement was used.

To improve presentation of the data, we published the transcriptomics and metabolomics metadata on online repositories (MetaboLights and GEO). Additionally, we provide all source data in supplementary tables 1 and 2, as well as metabolomics reporting standards in table S10. Metabolite dynamics are shown in Figure S2. We added this information in the revised manuscript:

“Gene expression data that support the findings of this study have been deposited in NCBI’s Gene Expression Omnibus with the accession code GSE131992 (<https://www.ncbi.nlm.nih.gov/geo/query/acc.cgi?acc=GSE131992>). Metabolomics data that support the findings of this study have been deposited in MetaboLights database with the accession codes MTBLS1044 (<https://www.ebi.ac.uk/metabolights/MTBLS1044>). The source data of Figures 1a-b, 2, 3a-c, 4b-c and 5b and Supplementary Figures 1, 3, 4, 5, 7 and 9 are provided as a Source Data file.”

Regarding the two replicates per time point: We decided to sample with higher frequency and reduce the number of replicates to 2. The reason is that a continuous and smooth time course informs also about precision and quality of the data (in addition to the two replicates). Four transcriptome samples had low RNA quality as indicated by Bioanalyzer measurements and we removed them from the analysis.

Below I get into more detail of these major issues, and finally add a list of minor concerns.

Limits to the method and validation.

The E.coli network is estimated to comprise around 300 TFs, but they input only 209 in the network component analysis (NCA). Is NCA limited only to TFs with a priori known target genes? If so, this is a major limitation to apply this approach in other genomes, where the regulatory network is much less characterized, or essentially unknown.

We obtained the transcription regulation network from the RegulonDB and Ecocyc data bases, which covers 223 transcription factors. For 209 regulators, the transcriptomics data covered at least one gene of the regulon.

NCA is indeed limited to TFs with *a priori* knowledge about the transcriptional regulatory network of the organism. In the revised text we discuss this point and alternative approaches to calculate transcription factors activities without a transcription regulation network:

“Here, we inferred TF activities with the NCA algorithm that requires a well-mapped transcription regulation network. While the transcription regulation network is known in E. coli, it is unknown for most other organisms. To overcome the need for a known transcription regulation network, the TF activities could be inferred from the transcriptome data directly without using prior knowledge about the transcription regulation network. Previous studies showed for example that machine learning methods can infer TF activities in E. coli based on transcriptomics data³⁸, and inference of regulatory metabolites with such methods was also suggested³⁹. Future approaches could even consider determining TF activities and regulatory metabolites simultaneously.”

I wonder if the dynamics of autoregulatory interactions, rather common in TFs, may pose difficulties to the reverse engineering within the NCA method.

Autoregulation of a transcription factor by itself is not a problem for the NCA algorithm. The transcriptome data covered expression of the transcription factors and the autoregulatory interactions are included in the transcription regulation network. Thus, the expression of the gene encoding the transcription factor is used to calculate the activity of the respective transcription factor like any gene in the regulon.

Out of 209, after filtering, the final result is the identification of potential allosteric effectors for 65 TFs. It is almost logically mandatory to discuss somewhere, what alternative are there to identify allosteric metabolites for the remaining 144 TFs.

We agree with this point and discuss in the revised text how interactions of the remaining TFs could be identified.

- Measurements of the exo-metabolome may identify regulators of two component systems.

“In our analysis, we excluded two-component systems, because they are likely responsive to external metabolites. By measuring the exo-metabolome it should be possible to identify effectors of two-component systems with the method proposed in this study.”

- Re-measuring of unprecise TF activities, e.g. with fluorescent transcriptional reporters:

“We also excluded TFs with poor estimates of activity profiles, and to include these TFs one could probe their activities with fluorescent transcriptional reporters as recently suggested¹⁶.”

- More specific perturbations to avoid correlations among metabolites:

“The main limitation in our study was that many metabolites showed similar dynamics, which in turn caused false positive predictions of metabolite-TF interactions. The high correlation among metabolites could be a general problem in metabolomics-based inference approaches²⁷. A solution for this problem is to enforce more specific metabolite concentration changes by localized perturbations of metabolism, for example by disturbing single enzymes. We anticipate that the transcriptome and metabolome of hundreds of locally perturbed metabolic states would provide sufficient information to faithfully map metabolite-TF interactions of an organism. An effective perturbation method is CRISPR interference, because of its potential to interfere with the expression of every enzymes of an organism.”

I do not see the logic to exclude two component systems. Many TFs have as signals external metabolites, so if that is the case of the two-component systems, it is not a unique feature of that class. Is NCA, or the experimental setting, that is limited to search for correlations with internal metabolites? This is relevant given the large fraction of external sensing of the regulatory network machinery.

We agree that external sensing is an important layer of cellular regulation. However, we did not measure external metabolites and therefore focus only on intracellular metabolites and their interaction with transcription factors. We describe this better in the revised text, and discuss how measurements of external metabolites could help to examine two-component-systems as well (see text above).

The approach relies on NCA, where transcription factor activity is an essential concept, nonetheless there is no explanation in the manuscript, how was TF activity obtained. It must have been inferred precisely using NCA, but this is said in a confusing way: “..we first calculated the activity of transcription factors and other regulators like sigma70 and sigmaS. To infer the activities of all transcriptional regulators in E.coli, we used NCA..”

We apologize that this part was confusing in the previous version and explain NCA in more detail in the revised manuscript:

“The relationship between transcriptional regulators and their target genes is well-characterized in E. coli, in the form of a transcription regulation network¹⁸. A well-mapped transcription regulation network allowed us to infer activities of transcriptional regulators from measured gene expression profiles using algorithms like Network Component Analysis (NCA)^{19,20}. The NCA algorithm estimates activity profiles of transcriptional regulators, which minimize the error between theoretical and measured gene expression profiles (for 2167 genes that are mapped to 209 transcriptional regulators in the E. coli transcription regulation network). In total we performed 100 searches with the NCA algorithm, each with a different randomized initial condition, such that we obtained means and confidence intervals for activity profiles of the 209 transcriptional regulators (Fig. 3a, Table S5).”

Additionally, we now provide on GitHub a well-documented Matlab code, which performs NCA using the transcriptome data from our Supplementary Excel Table.

In order to identify the TFs that correspond to the metabolites, the authors fitted Hill functions to all pairs of metabolites and TFs, generating 3990 potential interactions. These were then filtered, first eliminating all TFs that followed a simple on-off dynamics which reduced the potential interactions to close to 3000. Then metabolic feedback circuits were searched by calculation the distance in the metabolic network between the metabolite and the genes regulated by the TF. This brought potential interactions down to 336 and 65 TFs. There is absolutely no mention on how such a distance was measured.

We now apply a clearer distance criterion, which is based on the metabolic subsystems in the latest *E. coli* model, or whether a metabolite is a substrate/product of an enzyme in the regulon of the TF.

The distance criterion is described in the revised text:

“Instead, we reduced the number of putative interactions by using a distance criterion for metabolites and TFs: metabolite-TF pairs were only considered, if at least one target-gene of the TF encodes an enzyme that participates in the same metabolic subsystem as the metabolite or if the metabolite is a substrate or a product.”

This is also described in the methods section:

“The distance criterion for correlating metabolite-TF pairs (Fig. 4b) was also based on the genome scale model iJO1366²⁹. Pairs of metabolites and TFs were only considered if at least one of the two criteria was fulfilled. Criteria 1: the metabolite is a product or a substrate of an enzyme that is encoded by a target-gene of the TF. Criteria 2: the metabolite is listed in the same metabolic subsystem as an enzyme that is encoded by a target-gene of the TF. Subsystems of TFs were defined as the subsystems of the TF-targets in the genome scale model. Subsystems of metabolites were defined according to the Table S1.”

Second, I suggest the authors should show if this distance is indeed small for the known allosteric metabolites of the E.coli network, to strengthen their claim.

We thank the reviewer for this idea. The distance is indeed small for about 80% of the known interactions. We added this in the revised manuscript:

“We observed a similar proximity of metabolite-TF interactions in our literature network, because more than 80% of these interactions have a small distance in the E. coli genome-scale metabolic model²⁹ (Fig. S6).”

The revised methods part describes how the network of known metabolite-TF interactions was analyzed with the distance measure defined by Reznik *et al.* 2017:

“The distances of metabolite-TF interactions in the literature network (Fig. S6), were calculated as described recently⁴⁵. First, the distance of every gene to all metabolites was calculated by multiplying the stoichiometric matrix with the gene reaction matrix from the genome-scale metabolic iJO1366²⁹. This connectivity matrix was converted to a bipartite graph using an adjacency matrix. The minimal distance

of a transcription factor and a metabolite was then defined by the minimal number of steps between all genes in the respective regulon to every metabolite.”

I did not understand the addition of “a global generic regulator” to the connectivity matrix. Which global regulator is that? And what do we learn from this?

The global regulator accounts for the global activity of the RNA polymerase. This regulator is connected as an activator to every gene in the connectivity matrix. The global regulator is important, because it defines the basal activation of every gene. We added this information to the methods section:

“To account for global gene expression activity by the RNA polymerase we added a global regulator to the connectivity matrix. This global regulator was connected to all genes in the matrix and accounts for basal gene expression.”

The value of the data.

Table S6 has the correlation coefficient for Hill-type kinetics for all pairs of metabolites and TFs. I suggest the authors put in bold those coefficients that correspond to the 336 filtered interactions, given the quote to that table in the manuscript, and the relevance of their filtering. Additionally, a new table should be added listing the final set of identified potential TF-metabolite interactions.

We apologize that this was indeed difficult to understand in the previous version and modified supplementary Table S7. We now list the final set of 513 putative interactions and highlight the known and potentially new interactions.

More detailed observations.

The E.coli BW25113 strain was used. The authors did not mention if they checked consistency in the use of databases used for the known metabolites with this particular strain.

The databases used in this study are valid for BW25113. RegulonDB is valid for all *E. coli* K12 strains. EcoCyc includes information of MG1655 and BW25113. Metabolites are primary metabolites that are present in all *E. coli* strains.

In the first sentence of page 3, the authors refer to Fig 3b, when they mean Fig 3a.

Corrected

Figure S6 could be better changed the X scale to millimolar.

Corrected

The description in methods of the conditions of growth does not seem to me the most appropriate for the sake of easy comparison with other experiments in the literature. I think it should describe the molar

or millimolar concentration of each component, instead of describing the amounts added to 1 liter, such as “1 mL of 0.1M CaCl₂”.

Corrected

Finally, I should say that given the dependency on the NCA method, and other features of this work, it is my appreciation that it is not easy for a researcher to reproduce this work, or even the analysis from the raw data. I understand this is not a simple issue, I suggest the authors to think if there are means to enhance the reproducibility of the work somehow.

We provide on GitHub all Matlab code that performs Network Component Analysis and non-linear regression of metabolites and TF activities. All code uses source data in supplementary Excel tables as input.

Response to Reviewer 3

Lempp et al. propose a strategy for systematic identification of metabolites that control transcription through transcription factors by integrating dynamic metabolomics and transcriptomics measurements. The proposed method was used to both validate many known metabolite-transcription factor interactions in *E. coli* and to predict a significant number of potential new interactions. A small number of the new interactions were validated experimentally in vitro. The manuscript is overall well written and clear and the figures are informative.

Major comments:

1) The network component analysis (NCA) approach the authors use to infer transcription factor activities from global transcriptome profiles has been previously shown to be useful, but also suffers from some clear limitations including the requirement to a priori specify all the TF-target gene interactions that will be considered and the specific model of how multiple TFs acting on the same target promoter are assumed to contribute to the expression of the target gene. I have multiple suggestions regarding the NCA part:

a. The manuscript does not clearly state these underlying assumptions when the NCA results are described (page 3, middle paragraph). The major assumptions should be described here to inform the reader better of what NCA can and what it can not do.

We thank the reviewer for the helpful comments and suggestion for additional analysis. We agree that NCA relies on the information about the transcription regulatory network. In the revised text we provide more information about NCA:

In the results section:

*“The relationship between transcriptional regulators and their target genes is well-characterized in *E. coli*, in the form of a transcription regulation network¹⁸. A well-mapped transcription regulation network allowed us to infer activities of transcriptional regulators from measured gene expression profiles using algorithms like Network Component Analysis (NCA)^{19,20}. The NCA algorithm estimates activity profiles of transcriptional regulators, which minimize the error between theoretical and measured gene expression profiles (for 2167 genes that are mapped to 209 transcriptional regulators in the *E. coli* transcription regulation network). In total we performed 100 searches with the NCA algorithm, each with a different randomized initial condition, such that we obtained means and confidence intervals for activity profiles of the 209 transcriptional regulators (Fig. 3a, Table S5).”*

In the discussion:

*“Here, we inferred TF activities with the NCA algorithm that requires a well-mapped transcription regulation network. While the transcription regulation network is known in *E. coli*, it is unknown for most other organisms. To overcome the need for a known transcription regulation network, the TF activities could be inferred from the transcriptome data directly without using prior knowledge about the transcription regulation network. Previous studies showed for example that machine learning methods can infer TF activities in *E. coli* based on transcriptomics data³⁸, and inference of regulatory metabolites*

with such methods was also suggested³⁹. Future approaches could even consider determining TF activities and regulatory metabolites simultaneously.”

b. It would be informative to show the NCA inferred TF activities in Figure 1 in addition or instead of showing all the transcript levels as the TF activity data is actually what is used in all the other analyses.

We added a heatmap showing all transcription factor activities to Figure 3 of the revised manuscript.

c. If I understood it correctly the authors did not allow time lags between TF activity and transcription of its target genes (on the other hand they did consider time lags between metabolite levels and TF activity). Did the authors consider including time lags in the analysis or at least check whether considering time lags could have influenced the results?

The reviewer raises an important point, and we did indeed allow for a negative time-lag between transcription factor activities and the metabolite levels. This is now better described in the methods part and a new supplementary Figure S10:

“For each pair of metabolite and TF, we tested if a negative time-shift of the TF activity by one time point would improve the parameter estimation. This accounts for the fact that TF activities are derived from gene expression data, which could potentially succeed changes of metabolite levels (Fig. S10).”

The logic behind the time-lag is the following: TF activity changes immediately upon changes of the effector metabolite. However, because TF activity is based on RNAseq data we expect that the TF activity is lagging behind metabolite changes (due to synthesis of mRNA). NCA itself can not consider the time lag between expression of mRNA and the transcription factor activities. Therefore we allow the time-lag in the correlation analysis, and because there is no information about the exact delay of each transcript, we allowed time lags of either 0 or 1 data point.

2) As the authors themselves clearly state, the major weakness of the proposed approach is its high false positive metabolite-TF interaction prediction rate. This appears to be primarily driven by the specific experimental conditions used in the study where many metabolites and transcripts follow switch-like on-off dynamics as a function of the switch between non-starved and starved conditions. This forces the authors to filter the list of potential interactions using multiple criteria some of which are better justified than others. I have again several suggestions for clarifying/improving this part:

We thank the reviewer for suggesting these three additional analyses. The results are described below.

a. The authors a priori exclude two component systems from the analysis as these are not supposed to be modulated by internal metabolite levels. However, it would have been interesting to see as a kind of a negative control how many metabolite-TF interactions would have been inferred for two component systems if they were included and if they indeed would have had fewer inferred interactions than other types of TFs. These types of controls are important for determining how false positive prone the overall approach is.

We compared how metabolites correlate with transcription factors and two component systems. As shown in the Figure below, there was no difference for TFs and two component systems. Also the number of correlating metabolites ($R^2 > 0.75$) per TF and per two component systems is almost the same (25 metabolites). We believe the reason is that we did this analysis before filtering with the distance criterion. After the distance filter there are too few correlations left for 15 two component to make an actual point. We therefore did not include this analysis in the manuscript.

Correlation coefficient of 123 metabolites with 125 transcription factors (TF) and two component systems (TCS). Number of correlating metabolites per TF and per TCS. In average 25 metabolites correlate with TFs as well as TCS.

b. The second exclusion criterion the authors used was to exclude any TF that significantly correlated with simple on-off pattern (depending on the starvation condition). It could be interesting to try a different approach to this where specific metabolite-TF interactions would be included only if the metabolite-TF correlation was higher than the on-off pattern correlation.

The suggested filter is an interesting alternative but removes slightly more metabolite-TF interactions, which results in a final list of 430 interactions instead of 513. We decided to use the original filter.

c. The strongest exclusion criterion used by the authors was requiring that the metabolite and TF target genes are part of the same subsystem and the distance from metabolite to a target gene of the TF is less than two steps in the metabolic network. This step is not described well enough in the methods (i.e. what is the definition of a subsystem and what is meant by a step exactly). Also, it would be interesting to show the R^2 in Fig. 4a as a function of the metabolic distance used as a filtering criterion before the filtering (i.e. considering distances larger than 2 also). Finally, the authors should provide detailed information in supplementary material on exactly how many interactions were excluded based on the subsystem and distance criteria for each subsystem.

We now apply a clearer distance criterion and added a supplementary Figure S8 that shows how many TFs and interactions were excluded. The distance criterion is described in the revised methods section:

“The distance criterion for correlating metabolite-TF pairs (Fig. 4b) was also based on the genome scale model iJO1366²⁹. Pairs of metabolites and TFs were only considered if at least one of the two criteria was fulfilled. Criteria 1: the metabolite is a product or a substrate of an enzyme that is encoded by a target-gene of the TF. Criteria 2: the metabolite is listed in the same metabolic subsystem as an enzyme that is encoded by a target-gene of the TF. Subsystems of TFs were defined as the subsystems of the TF-targets in the genome scale model. Subsystems of metabolites were defined according to the Table S1.”

To better justify the distance criterion, we show that the distance of most know metabolite-TF interactions in *E. coli* is small:

“We observed a similar proximity of metabolite-TF interactions in our literature network, because more than 80% of these interactions have a small distance in the E. coli genome-scale metabolic model²⁹ (Fig. S6).”

As suggested by the reviewer we calculated the distance for all 15,375 interactions between 125 TFs and 123 metabolites. There was only a slight trend that with smaller distance the percentage of metabolite-TF pairs with correlations above $R^2 > 0.75$ increases (see Figure below). However, we did not include this in the manuscript because the effect is small and we anyway follow-up on putative interactions that have a small distance.

Fraction of metabolite-TF pairs that have $R^2 > 0.75$ at different distances (15,372 interactions between 125 TFs and 123 metabolites).

Minor comments:

1) Some additional interesting controls to check:

a. Did the energy charge correlated with any TF activity?

Yes, the energy charge correlated with almost the same TFs like ATP (75%).

b. Did the “global regulator” level correlate with any metabolite level?

The global regulator accounts for the global activity of the RNA polymerase. This regulator is connected as an activator to every gene in the connectivity matrix. However the global regulator relatively constant, which is caused by the normalized TPM values as input for the NCA. Nevertheless, the global regulator is important, because it defines the basal activation of every gene.

“To account for global gene expression activity by the RNA polymerase we added a global regulator to the connectivity matrix. This global regulator was connected to all genes in the matrix and accounts for basal gene expression.”

2) Page 8. Normalizing by TPM is not considered to be terribly good approach nowadays. Consider using alternative better normalization methods in future studies and potentially also check for this study whether there is any significant effect due to the particular normalization used. There are many good assessment studies of different RNA-seq normalization methods. This could also improve the reproducibility of the data.

As described in the point above, any normalization method for the RNAseq mainly influence the global regulator. However, we agree that in future studies other normalization methods for the RNAseq could be tested, to see if they improve the calculation of the network component analysis or the correlation results.

3) Page 8. The authors state that the Hill coefficients were “constrained to 10” – does this mean that they were constrained to be less than 10.

This statement was indeed unclear. It is correct that we used 10 as the upper boarder for the Hill coefficient in the Hill-type kinetic fit. We clarify this in the text:

“The Hill coefficient h was constrained to an upper value of 10.”

4) Page 9. “puffer” -> “buffer”

Corrected

REVIEWERS' COMMENTS:

Reviewer #2 (Remarks to the Author):

This is an improved manuscript with all considerations taken into account.
I consider major concerns have been addressed by the authors.

There is a minor, almost stylistic detail. A definition does not use the same word in it. And you have a sentence saying:

"Subsystems of TFs were defined as the subsystems of the TF-targets .."

I suggest you change the second "subsystems" word by another one that may help better understanding.

Reviewer #3 (Remarks to the Author):

The authors have addressed my and the other reviewers comments very thoroughly. They have also made all the data and code available as required.